# Large Scale Optical Projection Tomography without the Use of Refractive-Index-Matching Liquid

**DOI:** 10.3390/s23249814

**Published:** 2023-12-14

**Authors:** Petros Ioannis Stavroulakis, Theodore Ganetsos, Xenophon Zabulis

**Affiliations:** 1Institute of Electronic Structure and Laser, Foundation for Research and Technology—Hellas (FORTH), 700 13 Heraklion, Greece; 2Non-Destructive Techniques Laboratory, University of West Attica, 122 41 Egaleo, Greece; ganetsos@uniwa.gr; 3Institute of Computer Science (ICS), Foundation for Research and Technology—Hellas (FORTH), 700 13 Heraklion, Greece; zabulis@ics.forth.gr

**Keywords:** 3D reconstruction, 3D scanning, glass, plastic, transparent objects, optical projection tomography, non-adaptive refractive index matching liquid

## Abstract

The practical, rapid, and accurate optical 3D reconstruction of transparent objects with contemporary non-contact optical techniques, has been an open challenge in the field of optical metrology. The combination of refraction, reflection, and transmission in transparent objects makes it very hard to use common off-the-shelf 3D reconstruction solutions to accurately reconstruct transparent objects in three dimensions without completely coating the object with an opaque material. We demonstrate in this work that a specific class of transparent objects can indeed be reconstructed without the use of opaque spray coatings, via Optical Projection Tomography (OPT). Particularly, the 3D reconstruction of large thin-walled hollow transparent objects can be achieved via OPT, without the use of refractive-index-matching liquid, accurately enough for use in both cultural heritage and beverage packaging industry applications. We compare 3D reconstructions of our proposed OPT method to those achieved by an industrial-grade 3D scanner and report average shape differences of ±0.34 mm for ‘shelled’ hollow objects and ±0.92 mm for ‘non-shelled’ hollow objects. A disadvantage of using OPT, which was noticed on the thicker ‘non-shelled’ hollow objects, as opposed to the ‘shelled’ hollow objects, was that it induced partial filling of hollow areas and the deformation of embossed features.

## 1. Introduction

The accurate and practical optical 3D reconstruction of transparent objects has been an open challenge for the field of optical metrology [1,2,3]. The main difficulty in using conventional practical optical metrology tools such as structured light scanning, laser scanning, and photogrammetry to reconstruct transparent objects, is due to the combined phenomena of transmission, reflection, and refraction noticed in transparent objects [4]. This fact does not allow the use of the aforementioned practical optical metrology tools, all of which require high levels of diffuse surface reflection to operate [5].

The typical way of getting around this limitation is to render the transparent object’s surface opaque, by spray-coating it with a diffuse reflection layer [6]. The transparent object can then be reconstructed with conventional visible-spectrum optical metrology tools [6]. The process of spraying the object, however, is time-consuming, adds considerable costs to the reconstruction process, and might not be allowed in some applications (sensitive cultural heritage objects, food and drink containers) due to the risk of contamination.

There have been multiple attempts to create alternative spray-less digitization techniques for transparent objects in the past [1,2]. The proposed techniques include the use of OPT with refractive index-matching liquid [7], the fringe projection for detection of pattern differences [8], infra-red (IR) to detect induced surface heating [9,10], Infrared Digital Holography [11], ultra-violet (UV) fluorescence [12,13], a combination of X-ray tomography and photogrammetry [14], shape from polarization [15], a combination of polarization imaging and inverse rendering [16], shape from interaction [17], the visual hull technique [18], various AI-based image processing methods [19,20] terahertz (THz) Imaging and tomography [21,22], passive single-pixel imaging [23], and edge estimation computer vision techniques [24].

None of the spray-less techniques suggested above, however, possess the combination of advantages of conventional visible-spectrum optical metrology tools, that of being concurrently cheap-to-use, rapid, practical, non-contact, and easily automated.

In this work, we investigate the possibility of achieving the 3D reconstruction of large thin-walled transparent objects for quality control purposes in the beverage packaging industry (worth an estimated USD 144.40 billion in 2023 according to Ref. [25]) and for digital preservation purposes in cultural heritage applications. The solution proposed herein retains all the aforementioned characteristics of conventional optical metrology tools already in use in these industries today, but can additionally operate without the need to use opaque spray. To achieve this, the use of OPT [26] without the use of refractive index-matching liquid was investigated.

## 2. Materials and Methods

The novelty in our approach lies in the specific setup being able to take advantage of the cone-beam xCT principle, the Radon Transform, by swapping the detector and light source in the original fan beam architecture. Additionally, we carefully selected the objects to be thin-walled, cylindrically symmetric objects in air, in order to enable the measurement without the use of index-matching liquid. As shown in Appendices Section A.1 and Section A.2, the light propagation characteristics of visible light rays for these types of objects are close to what OPT requires in order to perform an accurate reconstruction. Some errors are still expected due to the minimal but non-zero refraction induced by the object’s sidewalls.

### 2.1. Software and Hardware Parameter Settings

The software used to perform the tomography calculations from the collected images was the Astra Software Toolbox v2.0 [27] a popular open-source X-ray CT package. The Astra toolbox was selected because it was developed by an academic team, it is a well-known and commonly used tool in FBP as documented in multiple academic papers on the subject, and therefore evaluated thoroughly in terms of accuracy and correctness of the algorithms used [27].

The available scanning setup configurations in the Toolbox are: for 2D scanning, a ‘fan-beam’ and a ‘parallel beam’ setup. For 3D scanning, ‘cone’ and ‘parallel3d’ setups are available. A diagram depicting all four readily available configuration modes is shown in Figure 1.

In this work, the ‘cone’ configuration Figure 1d was used by replacing the X-ray source with an optical camera, and the X-ray detectors with a field light source (LCD Panel). It is assumed that the camera used can be modeled by the ‘pinhole camera model’, and therefore we essentially use the 3D ‘cone’ beam geometry in reverse. For this reason, the calibration parameters required for this setup are not explicitly transferable to the optical camera setup in the sense that the result will not be of the correct scale.

In the ‘cone’ beam setup used, the following parameters are required to be set in the Astra Toolbox namely: CCD x and y pixel distance (set to 1), number of pixel rows in the detector (512), number of pixel columns in the detector (512), explicit projection angles (64 points of view), the distance between source and center of rotation (70 cm), the distance between the center of rotation and detector array (20 cm).

By using these values we acquire the correct object shape, but it is scale-less. Therefore, here we need to scale the object appropriately by either measuring its exact distance from the camera or by scaling the results with the measurement of a known object. In our case, we performed the latter.

A polished glass ball 80 mm in diameter (i.e., a ‘lens ball’ in specialist photography) was used for calibration, shown in Figure 2. Since the sphere is not hollow and has a convex shape, its outer surface was reconstructed by manually extracting its silhouette from each of the 64 axial rotations by the well-known ‘visual hull’ 3D reconstruction method [28]. The sphere’s reconstruction was then loaded in point cloud processing software (Cloud Compare [29], https://www.cloudcompare.org/ accessed on 7 March 2023) and its diameter was measured. The measured diameter was then divided by the true 80 mm sphere diameter to acquire the system’s calibrated scaling factor.

The computational hardware used for our calculations was a single laptop, equipped with a single Intel(R) Core(TM) i7-1065G7 CPU @ 1.30 GHz 1.50 GHz CPU and an onboard NVidia graphics card NVIDIA(R) GeForce(R) MX250 with 2 GB GDDR5 memory. However, for the Astra toolbox calculations, only the CPU mode was used without GPU acceleration. This means that the speed of the computation can be significantly improved from the current time required (2–5 min per object) by the use of a stronger CPU and GPU and by enabling GPU acceleration.

### 2.2. Reconstruction Algorithm Used

The Astra Toolbox software uses the filtered backpropagation algorithm [30] used in Xray-CT to reconstruct the measured volume density from the photographs taken at each rotation angle. The density of each voxel g(x,y,z) was calculated per slice f(x,y) at a particular *z* height by the integral in Equation (Equation 1) [27].
(1)$$f_{fbp}=\int q_{\theta }\left(xcos\theta +ysin\theta \right)d\theta$$
where θ is the rotation stage’s angle, x and y are the particular slice’s voxel locations and $q_{\theta }\left(t\right)$ is the filtered Fourier transform of the detected image described in Equation (Equation 2).
(2)$$q_{\theta }\left(t\right)=\int P_{\theta }\left(\omega \right)\left|\omega \right|e^{i2\pi \omega t}d\omega$$
where ω is the frequency in the Fourier domain, the spatial dimension of the 1D absorption measurement of each slice (the row of pixels of the photograph acquired at each rotation).

## 3. Results

### 3.1. Experimental Procedure

To measure the shape of thin-walled transparent items with OPT without the use of refractive index-matching fluid, the following experimental sequence, similar to that of a typical OPT workflow, was used:A rotation stage, a camera, and a field light source were set up as in the apparatus shown in Figure 3;Light from a field source (LCD Panel) was projected through the object and registered at the camera;The object is placed in the middle of the rotation stage and is rotated to acquire 64 rotational views around the object;The images acquired by the black and white camera are inverted so that areas of high absorption are bright and areas of low absorption appear darker;The images were processed by the Astra Toolbox’ [27] X-ray CT reconstruction software into a density volume;Thresholding the voxels of the 3D density volume to remove the low density of air. We are then left with the higher density voxels of the object;To extract a single surface from the thresholded density volume, we then post-process the object density volume slice-by-slice and line-by-line, to extract only the peak densities on each row of the image plane. Identifying the peaks represents the areas with the most dense material and hence those of the sidewall (Figure 4);The peak locations are then scaled using the calibrated scaling factor calculated in Section 2.1 and saved in a file in point cloud format.

### 3.2. Objects Selected

The types of objects to which this measurement principle is most suited to, are hollow, thin-walled, cylindrically symmetric objects, which do not induce considerable refraction as light traverses through them.

Hollow objects can be further subdivided into two categories, namely ‘shelled’ (objects whose internal and external surfaces are identical in shape and one is scaled down relative to the other by the size of the wall thickness) usually made of plastic. The other type is that of ‘non-shelled’ hollow objects (objects whose internal and external surfaces are not identical in shape and therefore have variations of thickness around the object), which are commonly made of glass.

Both types are of interest in manufacturing and cultural heritage applications, which this investigation aims to focus on. To test the category of ‘shelled’ objects that are commonly used in the beverage industry (e.g., soda, water bottles), we selected a soda bottle and two water bottles (Figure 5a–c). In order to test the category of hollow ‘non-shelled’ objects, applicable mainly to cultural heritage and drink and food containers made of glass, contemporary glass cups (two liqueur and one wine glass) with and without embossed features were measured. (Figure 5d–f).

However ‘non-shelled’ hollow objects are more difficult to reconstruct since the thickness of the material is not consistent around the whole object and therefore of varying refraction. They contain areas where the light passing through the object encounters thick layers of material and therefore gets refracted significantly (neck, base, bottom of cup area). Additionally, a lot of cultural heritage items also contain embossed features which further add to the variation of material thickness in specific areas of the object. We nevertheless tested such items, to test the limits of the suggested OPT method.

Before reconstructing the hollow ‘non-shelled’ selected in this work, they were cut down for two reasons. The first is that their sidewall thickness needed to be accurately measured by use of electronic calipers (Table 1) and this could not be done from the mouth area, which is much thicker. The second reason is so that they could fit in the field of view of the camera used in the OPT setup, which could only measure objects of about 150 mm in height (Figure 2). To measure the hollow areas of the glass objects selected (wine and liqueur glasses cup area), they were placed inverted onto the rotation table with the hollow side down and the stem and base pointing up. In any case, only the hollow parts of the glass items were considered, and the stem and base which contain thick material areas by default were ignored in this study.

## 4. Results

### 4.1. 3D Reconstruction Accuracy

To obtain reference 3D reconstruction results for the outside shapes of the measured items, we scanned the objects with a conventional white light structured light optical scanner used for industrial purposes, called the Shining 3D Einscan Pro 2X scanner (Figure 6). The reference scanner was calibrated to an accuracy of ±22 μm using the calibration plates, which are provided by Shining 3D. To use this scanner, the transparent objects needed to be coated with the ‘AESUB blue’ opaque spray coating.

Then, using the CloudCompare [29] point cloud software, both the reference point cloud reconstructions and the point cloud reconstructions created by the OPT process were first aligned by hand and then aligned more accurately via Iterative Closest Point (ICP) to an error tolerance of 10${}^{-4}$. Finally, to extract the dimensional error, the residual point cloud distances were calculated. The point cloud errors are depicted as color textures on the OPT point clouds in Figure 7 and the numerical average of the point cloud distances for each object is reported in Table 2.

There are multiple methods of comparing point clouds [31,32,33], using point-to-point, point-to-mesh, and mesh-to-mesh strategies. We opted for using the closest point-to-point distance, rather than comparing point-to-mesh or mesh-to-mesh, because in our case, the reference point clouds created by the structured light scanner were extremely dense, and therefore it was not necessary to create a mesh surface to accurately compare the point clouds, as suggested by Ref. [32].

For hollow ‘non-shelled’ objects, the average distance errors (±0.92 mm) as expected were higher on average than that measured for the hollow ‘shelled’ objects (±0.34 mm). This fact alone, however, was not reflective of the much wider type of errors experienced on these objects due to intense refraction effects, which manifested as distorted embossed shapes (Figure 5e), artificial ‘ghost material’ partially filling up the hollow areas (Figure 8), and the reduction the object’s size (Figure 8).

The maximum precision expected from the specific OPT setup, in general, was calculated by dividing the available camera pixels by the field of view and was found to be 0.5 mm per pixel. Therefore the minimum dimensional error expected on the lateral and vertical distances is half this value, ±0.2 mm. This sanity check is in line with the measurements we collected (Table 2). The measurement with the lowest error achieved was an average point cloud distance of ±0.3 mm between the OPT and the reference reconstructions for the ‘Selinari’ water bottle (Table 2).

### 4.2. Benefits over Visual Hull 3D Reconstruction

In this work, we use OPT to extract the shape of thin-walled objects as a single surface, and it is, therefore, worth qualitatively comparing it to another technique, the visual hull, which is very similar and used during calibration. The visual hull technique can operate in the visible spectrum without the use of spray coatings. It is well known that it can extract only the external convex shape via the use of silhouettes [28], which is why it was used to measure our calibration sphere in Section 2.1.

For hollow non-convex objects, however, it is known that this technique cannot be used as it produces a solid convex 3D shell around the object. For example, when this technique was used in Ref. [18] to reconstruct a wine glass, the opening of the hollow end was covered. Similarly, it is known that any convex cavities (small craters) around the external surface of the object are ‘filled up’ due to the nature of the visual hull technique. OPT being a tomography technique, similar to X-ray CT, does not have these drawbacks as it can reconstruct hollow objects and can also deal with convex surface structures.

What is more, OPT can also measure internal surfaces. To demonstrate OPT’s ability to measure internally, we placed two cut-offs of plastic bottles, one inside the other, and the reconstructed result is shown in Figure 9. However, since we could not perform a reference measurement for internal surfaces (e.g., using an X-ray CT machine), it was not possible to confirm the achievable accuracy.

### 4.3. Comparison with Multi-View Stereo (MVS) Photogrammetry

MVS requires a surface texture to operate, which is why it is well known to perform very poorly on transparent objects [17]. We attempted to reconstruct the objects in this study using MVS as it is one of the most commonly used camera-based reconstruction techniques today, in order to contrast its reconstruction quality to that achieved by the OPT technique.

In Table 2 the prohibitive RMS errors involved in reconstructing transparent objects using MVS, when compared to our reference 3D reconstruction performed via structured light can be noticed. In Figure 10, these large errors are visualized by aligning the point clouds to the reference reconstructions, and coloring each point of the MVS reconstructed the point cloud with its minimum distance to the reference point cloud.

### 4.4. Comparison with Neural Radiance Fields (NeRF)

NeRFs are a relatively new reconstruction technique [34]. It uses artificial intelligence to build a non-linear relationship between the input, which is a single continuous 5D coordinate (the spatial location (x,y,z) and viewing direction (θ,ϕ)), and the output, which is is the volume density and view-dependent emitted radiance at that spatial location.

It is primarily used for rendering purposes but it can also recover the voxelized 3D shape of the object. Due to the complication of light transport between views the reconstruction of transparent objects is not fully successful. It is however more successful than the multi-view stereo shown in Section 4.3.

In this section the point clouds created via NeRF with the data acquired by the reference reconstructions in Figure 11 are compared. It can be observed clearly in Figure 11 that NeRF perfoms better than MVS but worse than OPT. The numerical averages of the errors shown in Table 2 also confirm this observation.

### 4.5. Summary

A summary of the comparisons performed to our reference 3D reconstructions, is found in Table 2 where we numerically compare the average point cloud error achieved by OPT, MVS, and NeRF. It is clearly seen that OPT is much more accurate in reconstructing the external surface of these transparent objects most of the time, with its accuracy being an order of magnitude better than the other two.

If we perform a qualitative study in the 3D reconstruction quality of the same reconstructed object, shown in Figure 12, we notice that OPT retains the most surface details and also has the highest level of reconstruction completeness. Secondly, we notice NeRF with an acceptable level of completeness but without the ability to reconstruct any of the surface details, and lastly, MVS, which has both very poor completeness and reconstruction fidelity.

Compared to other reconstruction methods which have been suggested for the reconstruction of transparent objects mentioned in the introduction, the cost of OPT is minimal, as the only two things required are a field illumination source such as a large LED, a means of rotation, and a black and white camera. The speed of the method has to be divided into acquisition speed and data processing speed, which can be done asynchronously if required. Since the acquisition is performed by cameras, it can potentially be performed on the level of milliseconds. The data processing speed demonstrated here can also be improved many times over by the use of parallel GPUs and with the use of more professional hardware. Regarding the potential for complete automation, it can be completely automated either by adding a robotic arm or via the use of a conveyor belt.

When it comes to the specific class of transparent objects considered in this study, OPT therefore does seem to have the potential to provide near real-time 3D reconstruction at an incredibly low price and with much higher accuracy than any of the techniques that have preceded it so far.

## 5. Discussion

The use of OPT over traditional X-ray CT has many benefits. X-ray CT reconstructions are cumbersome, slow, expensive, and present health risks to the operators. On the other hand, one of the main downsides of OPT is the necessary use of index-matching liquid. This work demonstrates the use of OPT without the need to use index-matching liquid by successfully reconstructing a specific class of large hollow and thin-walled objects.

The use of OPT in this work was investigated for two use cases in particular: the quality control of plastic bottles in the beverage packaging industry, and the reconstruction of glass objects in the context of a cultural heritage digital preservation application. Representative plastic and glass objects for these cases were collected and reconstructed in 3D using OPT.

It was shown that for plastic bottles produced by the beverage packaging industry, an average accuracy of ±0.34 mm can be achieved with the setup used. The fact that the best point cloud accuracy achieved ±0.29 mm was close to the theoretical precision of the setup ±0.21 mm indicates that with an even more precise setup, a higher accuracy could potentially be achieved.

For glass objects in the context of cultural heritage on the other hand, which on average have thicker sidewalls and also have some areas with thick optical paths, considerable refraction is produced. Areas that suffer more from this were the lower parts of the hollow areas, which have thicker sidewalls than the rim, the glass joint between the stem and the vessel, and the embossed designs on the glass surface. The average shape error of the glass objects measured was higher than that of plastic objects, as expected, at ±0.92 mm.

The relatively small numerical difference in average dimensional error of ≈0.6 mm between the ‘shelled’ and ‘non-shelled’ hollow objects however, does not accurately reflect the rather large qualitative difference of the reconstruction results achieved on the ‘non-shelled’ hollow objects, as it was noticed that hollow areas were being filled with ‘ghost material’, embossed features were being distorted, and objects were appearing smaller than their true size.

The dimensional errors experienced on ‘non-shelled’ hollow objects were in part expected as they are due to the non-use of refractive-index matching material. Not using ‘refractive index matching liquid’ induces a large amount of refraction for hollow objects with thick sidewalls, as predicted analytically in Section 2.

The advantages of using OPT for particular cultural heritage and industrial applications, where the use of opaque spray coatings is forbidden, is first and foremost that there is no need to use opaque spray coatings to perform the 3D measurement. A second great advantage is the ability to reconstruct internal structures (provided they too are thin-walled), something conventional optical tools cannot do. Additionally, it retains all the characteristics which make conventional optical metrology tools (structured light, laser scanning, and photogrammetry) attractive to industrial and cultural heritage applications, namely: it is cheap and easy to use, safe for human exposure, is camera-based, and it has the potential to perform extremely fast data acquisition, whilst also having a high degree of reproduction fidelity and accuracy.

The disadvantage of using OPT is that a narrow class of objects are possible to reconstruct with high accuracy, namely hollow ‘shelled’ objects such as plastic bottles. When the sidewalls of the object start to either become much larger in size or divert too much from being cylindrically symmetric, refraction effects become important enough to distort the shape of the objects considerably and therefore reduce the accuracy of the technique.

When comparing the results with other established 3D reconstruction techniques namely MVS and NeRF, we can see the qualitative difference both in completeness and surface feature detail. A side-by-side comparison of the point clouds created is shown in Figure 12.

## 6. Conclusions and Future Work

In summary, it is demonstrated to the best of our knowledge for the first time, that it is possible to use OPT without the use of refractive index-matching liquid for the reconstruction of objects with sizes larger than 10 mm. The only condition that needs to be met to achieve high accuracy is that the object’s sidewall must be thin and consistent enough not to induce significant amounts of refraction. This technique could therefore potentially be used for dimensional quality assurance of hollow ‘shelled’ objects such as plastic bottles in the beverage packaging industry. On the other hand, for the reconstruction of ‘non-shelled’ objects such as glass cultural heritage objects, the reconstruction technique is shown to be much more prone to errors due to the higher levels of refraction, which produce severe dimensional and aesthetic distortions to embossed features, as well as artificial object shrinking and the artificial filling of some parts of the hollow object’s volume.

In future work, it is aimed to improve the speed of the data processing at least 10-fold, by using improved hardware including fast GPU processing that is supported by Astra Toolbox. The accuracy of the technique is aimed to be improved to a wider class of objects, namely hollow “non-shelled” objects by mitigating the refraction errors induced via optical modeling of the light propagation through the object. Furthermore, the maximum shape deviation that can be tolerated will be simulated numerically via Zeemax (Canonsburg, PA, USA), an optical simulation software that performs ray tracing and is used in professional lens and camera design.

## 7. Patents

An international PCT patent application with number: PCT/GR2023/000051 has been submitted as a result of this work.

## Figures and Tables

**Figure 1 sensors-23-09814-f001:**
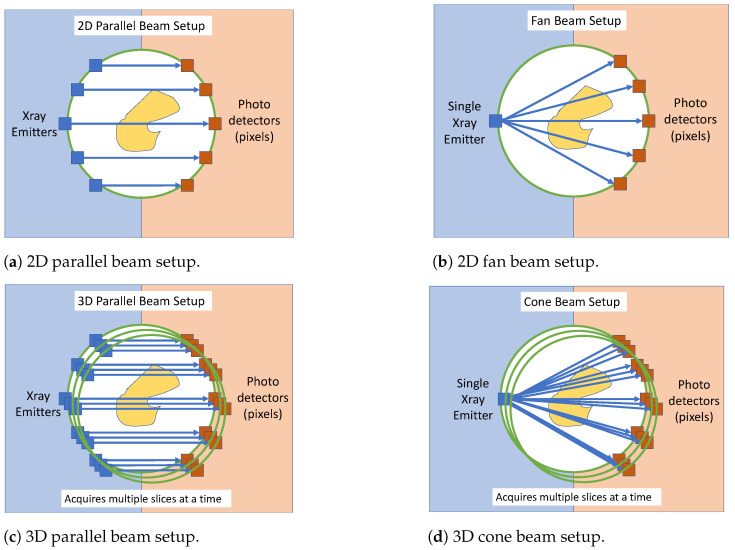
The four standard 3D tomography setups available in the Astra Toolbox X-ray tomography software [27]. (**a**) 2D parallel beam setup. (**b**) 2D fan beam setup. (**c**) 3D parallel beam setup. (**d**) 3D cone beam setup.

**Figure 2 sensors-23-09814-f002:**
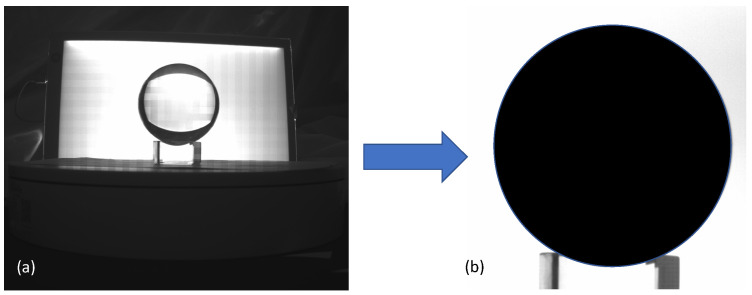
Back-illuminated images (**a**) from the setup camera was used to extract a reference solid black silhouette (**b**) and then reconstructed using the visual hull technique.

**Figure 3 sensors-23-09814-f003:**
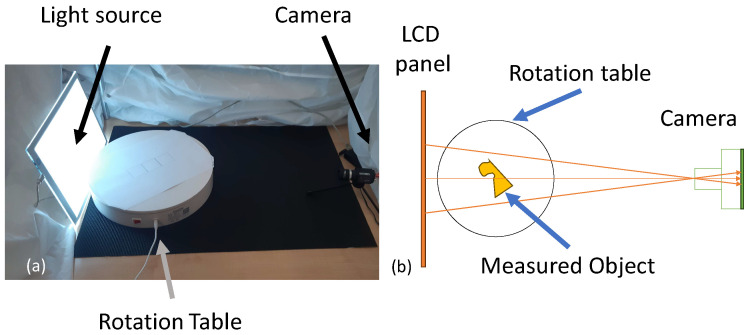
Experimental setup used for acquisition of object photographs consisting of a screen, a rotation table, and an industrial camera. (**a**) Photograph and (**b**) diagram.

**Figure 4 sensors-23-09814-f004:**
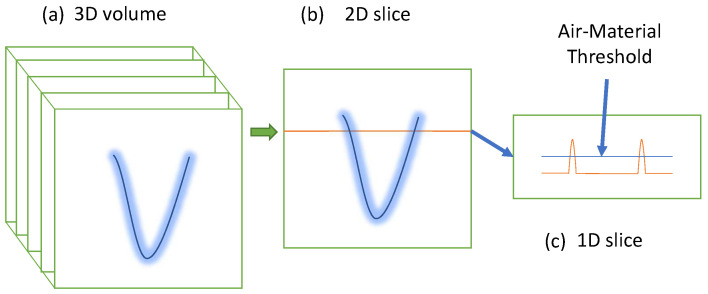
Process of thresholding (**a**) 3D voxel density to isolate transparent object surface. Each thresholded vertical voxel slice (**b**), is processed line-by-line (**c**), to extract the absorption peaks on each row.

**Figure 5 sensors-23-09814-f005:**
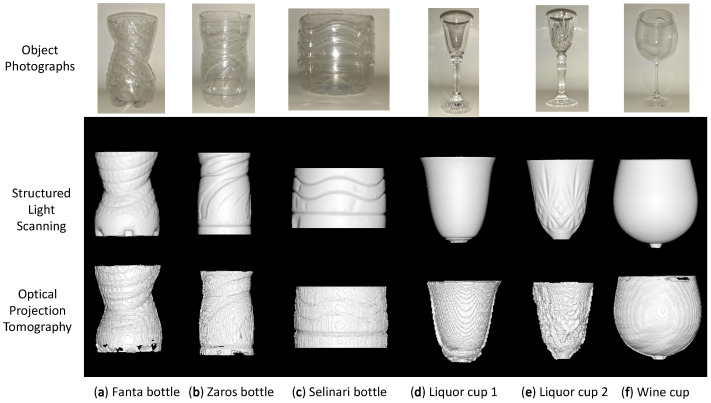
Photographs of transparent objects measured (top row), 3D reconstructions acquired from the “Shining 3D Pro 2X” industrial 3D scanner (middle row), and 3D reconstructions by our OPT method (bottom row). (**a**) Fanta bottle (**b**) Zaros bottle (**c**) Selinari bottle (**d**) Liquor cup 1 (**e**) Liquor cup 2 (**f**) Wine cup.

**Figure 6 sensors-23-09814-f006:**
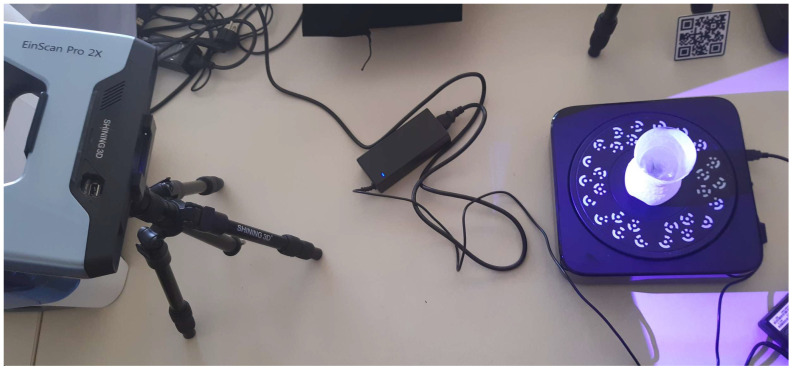
Static structured-light experimental setup used for acquisition of the reference 3D reconstructions.

**Figure 7 sensors-23-09814-f007:**
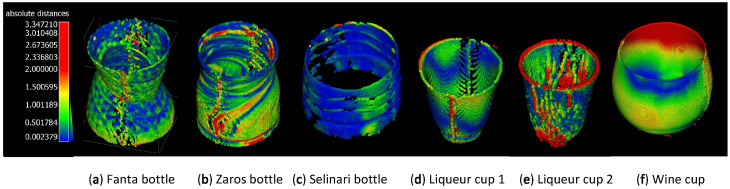
Use of color texturing on the OPT reconstructed point clouds of the objects in Figure 5, which visually represent the value of the closest distances of each point in the OPT reconstruction from the reference point cloud.

**Figure 8 sensors-23-09814-f008:**
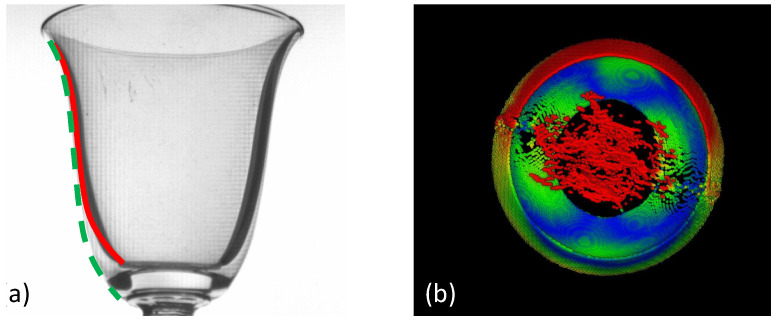
(**a**) Image from object in Figure 5, showing severely refracted liqueur glass outer edge (red line), compared to the actual surface edge (dashed green line), which manifests as an erroneous material density and therefore erroneous surface shape, (**b**) top view of wine glass 3D reconstruction in Figure 5f showing erroneously partially filled hollow area of the wine glass cavity.

**Figure 9 sensors-23-09814-f009:**
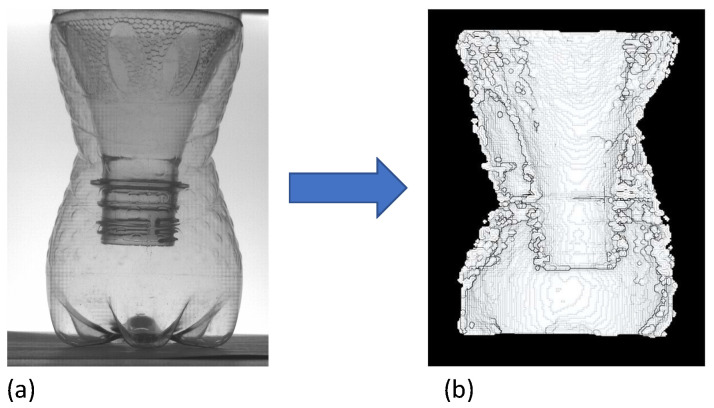
Demonstration of OPT’s ability to reconstruct internal surface structures, photo of a mixed bottle structure (**a**), and 3D reconstruction sliced in half (**b**), showing the internal structure.

**Figure 10 sensors-23-09814-f010:**
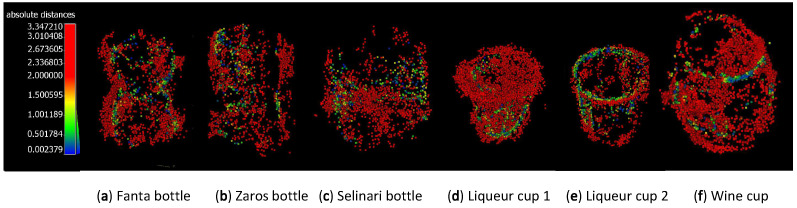
Use of color texturing on the MVS reconstructed point clouds of the objects in Figure 5, which visually represent the value of the closest distances of each point in the MVS reconstruction from the reference point cloud.

**Figure 11 sensors-23-09814-f011:**
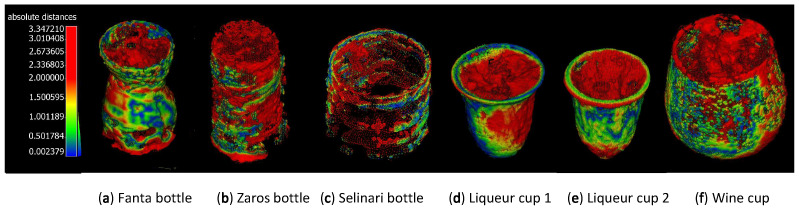
Use of color texturing on the NeRF reconstructed point clouds of the objects in Figure 5, which visually represent the value of the closest distances of each point in the MVS reconstruction from the reference point cloud.

**Figure 12 sensors-23-09814-f012:**
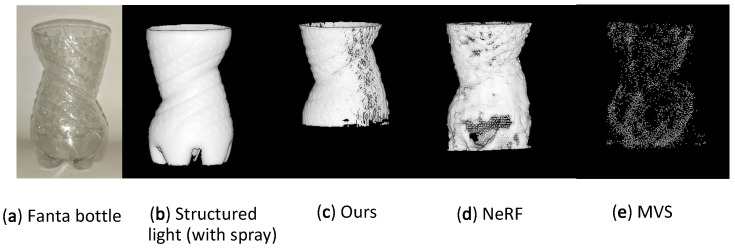
Qualitative comparison of different reconstruction methods results from left to right: (**a**) Object photograph, (**b**) structured light reconstruction, (**c**) OPT reconstruction (ours), (**d**) NeRF reconstruction, (**e**) MVS reconstruction.

**Table 1 sensors-23-09814-t001:** Thickness measurements of the plastic bottle sidewalls using electronic calipers.

Object	Average Thickness of Rim Sidewall (8 Measurement Average)
Fanta soda bottle	0.2 mm
Zaros water bottle	0.1 mm
Selinari large water bottle	0.1 mm
Liquor glass 1	2.1 mm
Liquor glass 2	2.3 mm
Wine glass	0.7 mm

**Table 2 sensors-23-09814-t002:** Results of comparing the point clouds acquired by MVS, NeRF, and OPT on the transparent objects shown in Figure 5 to the reference measurements acquired from an industrial-grade 3D scanner.

Object	Average Error Distance (MVS)	Average Error Distance (NeRF)	Average Error Distance (OPT/Ours)
Fanta soda bottle	6.1 mm	3.4 mm	0.3 mm
Zaros water bottle	6.8 mm	5.8 mm	0.4 mm
Selinari large water bottle	8.6 mm	3.2 mm	0.3 mm
Liquor glass 1	5.9 mm	3.7 mm	0.5 mm
Liquor glass 2	3.1 mm	2.2 mm	0.8 mm
Wine glass	10.4 mm	5.6 mm	1.5 mm

## Data Availability

The data related to this projects is not publicly available due to commercialization and PCT patent application embargo. They will be made available on request after a decision on the patent has been reached.

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
