# Peer review of "Large Scale Optical Projection Tomography without the Use of Refractive-Index-Matching Liquid"

_sensors, 2023, doi:10.3390/s23249814_

Round 1
Reviewer 1 Report
Comments and Suggestions for Authors
The precise reconstruction of transparent objects in three dimensions through optical means has long been challenging in optical metrology. The primary obstacle lies in the transmission, reflection, and refraction of light through these objects. In this study, the authors propose a novel large-scale optical projection tomography technique that eliminates the need for refractive index matching liquids, thereby addressing some limitations of previous methods. While this research is innovative and presents a viable solution to the problem of high-precision optical 3D reconstruction of transparent objects, several issues must be addressed before the manuscript can be accepted for publication.
1. Firstly, in the abstract, we recommend providing the full English name of Optical Projection Tomography (OPT) to ensure everything is clear and familiar with the term. Additionally, providing a brief explanation of OPT may be helpful to ensure clarity.
2. Secondly, we suggest adding "non-adaptive refractive index matching liquid" as a keyword, as this is a unique aspect of this work.
3. Thirdly, considering the current research on THz imaging in 3D reconstruction of transparent objects, it may be appropriate to include some related work in the introduction.
4. Fourthly, in section 3.2, we recommend clearly explaining the proposed reconstruction algorithm and how it differs from Fourier slicing. Furthermore, while the authors explain the use of Astra Toolbox Filtered Backprojection (FBP) algorithm, they should clarify how their work improves upon existing methods.
5. Astra Toolbox is a powerful MATLAB and Python toolbox that utilizes GPU acceleration for basic forward and back-projection operations. However, it is unclear whether GPUs were used in the experimental tests conducted by the authors. Therefore, we recommend clarifying whether GPU acceleration was utilized in the experiments. Additionally, the formula for the calibration scaling factor mentioned in section 3.1 is missing from the original text and should be provided.
6. Since commercial or open-source software was used extensively in the experimental tests, it is important to determine whether these tools significantly impacted the final 3D reconstruction of transparent objects. We suggest conducting comparative experiments to evaluate the effectiveness of the proposed method against other commonly used tools. Moreover, we recommend including phase imaging and binocular ranging control experiments in the reconstruction experiments to enhance their persuasive power. These control experiment groups could also help further validate the proposed method's effectiveness in solving the problem of high-precision optical three-dimensional reconstruction of transparent objects in industry settings.
In conclusion, while this study presents an innovative approach to solving the problem of high-precision optical 3D reconstruction of transparent objects, there are several issues that must be addressed before they can be considered for publication. By addressing these concerns and conducting additional experiments, we believe that this study has the potential to significantly contribute to the field of optical metrology.
Comments on the Quality of English LanguageThere is some room for improvement in English.
Reviewer 2 Report
Comments and Suggestions for Authors
The paper discusses the challenge of reconstructing transparent objects in three dimensions using optical metrology techniques. The main difficulty lies in the combined phenomena of transmission, reflection, and refraction in transparent objects, which make it hard to use conventional 3D reconstruction solutions without coating the object with an opaque material. The document proposes the use of Optical Projection Tomography (OPT) without the need for refractive-index-matching liquid to reconstruct large thin-walled hollow transparent objects accurately. The authors compare the 3D reconstructions achieved with their proposed OPT method to those obtained with an industrial-grade 3D scanner and report average shape differences for plastic and glass objects. Although the idea is interesting, it is recommended that the authors address the following points:
1. Are there any limitations or challenges associated with using the OPT method without the use of refractive index-matching liquid? How does this affect the accuracy or reliability of the measurements?
2. Can you provide more information about the resolution or accuracy of the reconstructed object shape? How does it compare to other 3D reconstruction methods?
3. Some works related to the 3D reconstruction reflection, and refraction in transparent objectscan be referred in the background or introduction part. For example, Cryst. Growth Des. 2023, 23, 11, 7992–8008, DOI: 10.1021/acs.cgd.3c00780.
4. Are there any plans for future research or improvements to the OPT method? Are there any limitations or areas that need further investigation?
5. How does the OPT method compare to other non-contact optical metrology tools in terms of cost, speed, and automation? What are the advantages and disadvantages of using OPT in manufacturing and cultural heritage applications?
6. The experimental setup and procedure are well-described, but it would be beneficial to include some images or diagrams to help visualize the setup.
7. It would be useful to provide more justification for why these objects were chosen and how they represent the types of objects that this measurement principle is most suited for.
8. Were there any specific applications or industries that were targeted for this research? How could the findings be applied in real-world scenarios?
Comments on the Quality of English LanguageProofreading service is encouraged.
Reviewer 3 Report
Comments and Suggestions for Authors
The manuscript has to be seriously improved. We are first of all doubtful regarding the practical importance and scientific soundness of the method. It can be applied only to a very narrow range of possible 3D objects. It is time consuming, but provides a rather modest or even low accuracy of +/- 1 mm. The authors are to comment on this important note. Then, the division of possible objects into plastic and glass ones seems strange. Today there is a great variety of plastics with different refraction indexes, so the predicted accuracy of the method has to be emphasized in terms of refraction index and thickness of the object walls. Next question. What is the maximal possible deviation from the circular shape which can be measured and what will happen with the accuracy for large deviations? Less important notes. Te consideration of the beam disposition by the plate is a problem for the beginners. If the paper will be accepted, then, to our opinion,this part has ti be strongly shortened or removed or transformed into some Appendix. Finally. The accuracy has to be represented in 1 or atmost 2 figures, i.e. instread of +/- 0.32, 0.85 or 1.47 we are to see +/- 0.3, 1.0 or 1.5 mm
Round 2
Reviewer 1 Report
Comments and Suggestions for Authors
The author has made careful and thorough revisions based on the review comments, and the revised manuscript presents the author's viewpoint well. The feature of this manuscript is that the topic of large-scale optical projection tomography imaging without the use of refractive index matching liquids has been enhanced to a certain extent.
In summary, this manuscript has reached the level of publication in your journal, and we recommend its official publication.
Comments on the Quality of English LanguageThere is still some room for improvement.
Reviewer 2 Report
Comments and Suggestions for Authors
publish
Reviewer 3 Report
Comments and Suggestions for Authors
The drawbacks were fixed and corrected. The paper is suitable for publication. I am still very doubtful about the scientific importance and soundness of the future paper, but still it can published as the original idea and proposal.